# Comparison of the Morpho-Physiological and Molecular Responses to Salinity and Alkalinity Stresses in Rice

**DOI:** 10.3390/plants13010060

**Published:** 2023-12-23

**Authors:** Abdelghany S. Shaban, Fatmah Ahmed Safhi, Marwa A. Fakhr, Rajat Pruthi, Mahmoud S. Abozahra, Amira M. El-Tahan, Prasanta K. Subudhi

**Affiliations:** 1School of Plant, Environmental, and Soil Sciences, Louisiana State University Agricultural Center, Baton Rouge, LA 70803, USA; rpruthi@agcenter.lsu.edu; 2Botany and Microbiology Department, Faculty of Science (Boys), Al-Azhar University, Cairo 11884, Egypt; mahmoudabozahra@azhar.edu.eg; 3Department of Biology, College of Science, Princess Nourah bint Abdulrahman University, Riyadh 11671, Saudi Arabia; faalsafhi@pnu.edu.sa; 4Botany Department, Faculty of Science, Fayoum University, Fayoum 63514, Egypt; maa29@fayoum.edu.eg; 5Green materials Technology Department, Environment and Natural Materials Research Institute, City of Scientific Research and Technological Applications (SRTA-City), Borg El-Arab, Alexandria 21934, Egypt; 6Plant Production Department, Arid Lands Cultivation Research Institute, City of Scientific Research and Technological Applications (SRTA-City), Borg El-Arab, Alexandria 21934, Egypt; aeltahan@srtcity.sci.eg

**Keywords:** *Oryza sativa*, gene expression, abiotic stress, salinity, alkalinity, genetic improvement

## Abstract

Rice is a major food crop that has a critical role in ensuring food security for the global population. However, major abiotic stresses such as salinity and alkalinity pose a major threat to rice farming worldwide. Compared with salinity stress, there is limited progress in elucidating the molecular mechanisms associated with alkalinity tolerance in rice. Since both stresses coexist in coastal and arid regions, unraveling of the underlying molecular mechanisms will help the breeding of high-yielding stress-tolerant rice varieties for these areas. This study examined the morpho-physiological and molecular response of four rice genotypes to both salinity and alkalinity stresses. Geumgangbyeo was highly tolerant and Mermentau was the least tolerant to both stresses, while Pokkali and Bengal were tolerant to only salinity and alkalinity stress, respectively. A set of salinity and alkalinity stress-responsive genes showed differential expression in the above rice genotypes under both stress conditions. The expression patterns were consistent with the observed morphological responses in these rice genotypes, suggesting the potential role of these genes in regulating tolerance to these abiotic stresses. Overall, this study suggested that divergence in response to alkalinity and salinity stresses among rice genotypes could be due to different molecular mechanisms conferring tolerance to each stress. In addition to providing a basis for further investigations into differentiating the molecular bases underlying tolerance, this study also emphasizes the possibilities of developing climate-resilient rice varieties using donors that are tolerant to both abiotic stresses.

## 1. Introduction

A variety of environmental stresses adversely influence plant growth, development, and productivity. Salinity, an important abiotic stressor, is caused by the presence of excessive amount of neutral salts in the soil. On the other hand, alkalinity results from the excess accumulation of bicarbonate and carbonate ions, leading to high soil pH and reduced availability of certain essential nutrients, such as iron, zinc, and manganese. Both stresses have diverse harmful impacts on both the morphology and physiology of plants, including disruptions to ion balance, photosynthesis, and water uptake. Such disruptions result in significant yield loss due to growth retardation and poor reproductive performance. To cope with these stresses, plants have developed several complex molecular mechanisms which include the regulation of ion transporters, production of osmo-protectants, and antioxidant systems.

Despite the importance of rice (*Oryza sativa* L.) in providing food security to the growing world population, its production is severely inhibited by salinity and alkalinity stresses. To address the challenges posed by saline and alkaline stresses in rice cultivation, it is essential to decipher the molecular and physiological basis of tolerance mechanisms in different rice genotypes. Several morphological traits have been identified as indicators of saline or alkaline tolerance in rice. For example, root length, shoot length, and biomass are commonly used to evaluate plant’s performance under stress conditions [1,2,3]. Leaf chlorophyll content is closely related to photosynthesis and stress tolerance in plants [4,5,6]. Hussain et al. and Zhang et al. [7,8] reported that salinity stress significantly reduced the plant height, tillers per plant, and grain yield. Likewise, Ma et al. (2022) observed a significant reduction in root length, biomass, and mineral nutrient uptake in rice plants under alkaline stress [9]. Numerous studies have been conducted to identify rice genotypes that exhibit tolerance to saline and alkaline conditions. Zhang et al. [8] demonstrated that some rice genotypes were more tolerant to alkalinity than others due to better root growth, photosynthetic activity, and ion balance.

There has been significant progress in untangling the molecular mechanisms that govern abiotic stress tolerance in rice. Many genes play significant roles in protecting rice plants against saline and alkaline stresses. The major mechanisms involved in seedling stage salinity tolerance are the maintenance of ion homeostasis and osmotic adjustment, whereas high pH tolerance is important for plants’ adaptation under alkaline environment [10]. An abundance of sodium and chloride ions in the soil can lead to an ionic imbalance by interfering with essential ion influx, particularly K^+^ [11]. Additionally, a high Na^+^ concentration reduces the water absorption capacity of roots, leading to a drought-like situation [12]. A number of genes belonging to HKT (High-affinity K^+^ transporter) family play a crucial role in the regulation of Na^+^ and K^+^ balance under saline conditions [13,14]; *OsHKT1;1* helps in reducing Na^+^ accumulation in shoots to tolerate salt stress [15,16]. The balance between Na^+^ and K^+^ ions is maintained by the SOS pathway genes or the HKT transporter genes, which exclude Na^+^ from the cytosol or selectively unload Na^+^ from xylem sap, respectively [17,18,19]. The genes associated with ion transport such as Na^+^/K^+^ antiporter *SOS1* (Salt Overly Sensitive 1), *NHX* (Na^+^/H^+^ antiporter), and *AKT1* (K^+^ channel) were significantly upregulated under salt stress [10]. On the other hand, several alkalinity tolerance genes related to ion homeostasis and oxidative stress include *OsHKT1; 5* [20,21], *OsPPa6* [22], and *OsY3IP1* [23]. High-throughput sequencing and transcriptomic analysis have revealed genes and pathways that are differentially expressed under both stress conditions [5,8,24]. The expression dynamics of genes related to Na^+^/K^+^ homeostasis have been studied in different rice cultivars under salinity stress [25,26]. The role of *OsHAK20* in imparting salinity tolerance is due to the maintenance of Na^+^/K^+^ homeostasis, while *STRK1* is suggested to enhance salinity tolerance at the seedling and flowering stages [27,28]. The increased proline accumulation and reactive oxygen species (ROS) scavenging capacity by *OsMADS25* enhanced tolerance to salinity stress [29].

The early transcriptomic response of rice to alkalinity stress indicated that detrimental effects involved genes associated with enzyme activity, biosynthesis, metabolism, and binding activity [30]. A transcriptomic study suggested that alkalinity-responsive genes at the early seedling stage were associated with hormone signal transduction and secondary metabolite biosynthesis [5]. Both QTL mapping and genome-wide association studies identified candidate genes for traits associated with alkali tolerance [31,32,33]. In alkaline soils, Fe deficiency is due to its insoluble hydroxide and oxide forms in soil under high-pH conditions [34]. Plants with increased uptake of Fe under alkaline conditions show high pH tolerance [35]. *OsIRO3*, a bHLH-type transcription factor that negatively regulates the Fe-deficiency response in rice, was associated with alkalinity tolerance in *japonica* rice [36]. *OsPPa6* is an important osmotic regulatory factor and *OsY3IP1* has a possible role in suppressing photooxidative damage under stress conditions in rice [22,23]. A glucan endo-1,3-beta-glucosidase precursor (LOC_Os09g32550) and an OsFBX335-F-box domain containing protein (LOC_Os09g32860) have been identified as negative regulators for alkalinity tolerance in rice [32], which was supported by the differential expressions of a glucan endo-1,3-beta-glucosidase gene in the root tissue under saline stress in rice [37] and reduced abiotic stress tolerance in rice due to overexpression of the F-box protein gene [38]. Another gene, *OsLOL5*, improved the saline–alkaline stress resistance of rice via the active oxygen detoxification pathway [39].

Although salinity and alkalinity stresses are caused by different types of salts, there are many similarities with regard to morpho-physiological responses in both stress conditions. There might be some similar mechanisms for tolerance response to both stresses. Since many genes associated with both salinity and alkalinity stress have been reported, the expression of those genes in a set of rice genotypes with varying level of salinity or alkalinity tolerance can provide some insights into the mechanisms associated with tolerance response to both stresses. We hypothesized that salinity stress-responsive genes differ in expression in salinity-tolerant genotypes compared with the alkalinity-tolerant rice genotypes and vice versa.

Despite the extensive research conducted on the responses of rice plants to salinity and alkalinity stresses, additional studies are needed to decode the molecular mechanisms and genes associated with tolerance to both stresses. Therefore, in this study, we compared the morpho-physiological responses of four rice genotypes with varying levels of salinity and alkalinity tolerance and compared the expression of a set of saline and alkaline stress-responsive genes under both stresses.

## 2. Results

### 2.1. The Performance of Rice Genotypes under Saline and Alkaline Stress

The four rice genotypes showed varying levels of tolerance to saline and alkaline stress based on measurements of different stress-responsive traits (Table 1 and Table 2). For morphological traits such as chlorophyll content, root length and shoot length, variations among the genotypes under both stresses were significant, but treatment effects were significant for shoot length (SL) and root length (RL) (Table 1 and Table 2; Appendix A). The genotype x treatment interaction was significant only for visual salt injury score (SIS), which is an important indicator for tolerance to saline and alkaline stresses. There was significant variation in SIS among the genotypes under both stresses. Under salinity stress, Pokkali and Geumgangbyeo recorded an SIS of 3.0, which is significantly different from the SISs of Bengal (6.0) and Mermentau (7.0). However, under alkaline stress, Geumgangbyeo and Bengal performed best, with SIS scores of 3.3 and 4.3, respectively, whereas they were higher for Pokkali (5.7) and Mermentau (7.7). There were significant differences in the SISs among the rice genotypes under different stress conditions. Specifically, the SIS in Mermentau was found to be statistically different from Geumgangbyeo and Bengal, and the SIS in Bengal between the two treatments was also different. There was a significant increase in the SIS of Pokkali under alkaline stress compared with saline stress (Figure 1 and Table 1 and Table 2).

Geumgangbyeo recorded the highest chlorophyll content, while Mermentau had the lowest chlorophyll content under both stresses. On the other hand, Pokkali had the maximum shoot and root growth under both stress conditions. Interestingly, the shoot length of Pokkali was lower under alkaline conditions than under saline conditions. A similar pattern was observed for Bengal and Mermentau. In contrast, Geumgangbyeo showed consistent shoot length under both alkaline and saline stresses. There was no significant difference in the shoot length in Geumgangbyeo under both alkaline and saline stresses (Figure 1).

Our study examined several physiological traits, including SNC, SKC, RNC, RKC, SNaK, and RNaK, in rice genotypes (Table 1 and Table 2; Figure 2). Analysis of variance revealed significant differences among genotypes for RNC, RKC, and RNaK under both stresses. However, the effect between both treatments was significant for SKC, RKC, SNaK, and RNaK (Appendix A). Genotype x treatments were significant for only SKC and SNaK. Each genotype had a higher SKC under alkaline stress than saline stress. Furthermore, our investigation detected notable variations in RNC and RKC among genotypes. Mermentau accumulated the highest concentration of Na^+^ in roots under alkaline conditions, followed by Pokkali, Bengal, and Geumgangbyeo. The results were different under saline stress with the lowest accumulation of Na^+^ in the roots of Pokkali and Geumgangbyeo, while Bengal and Mermentau had the highest Na^+^ accumulation in their roots. For root K^+^ content, Pokkali accumulated the highest K^+^ under alkaline stress, followed by Mermentau, Bengal, and Geumgangbyeo. However, the trend was different for root K^+^ accumulation under saline stress, with Bengal recording the maximum K^+^ followed by Pokkali, Geumgangbyeo, and Mermentau. There was no significant difference for SNaK among genotypes in both stresses and RNaK under alkaline stresses. Bengal recorded the lowest SNaK under both stress conditions, while Mermentau had a higher SNaK ratio under saline stress. The average SNaK and RNaK were higher under saline stress in comparison with alkaline stress.

### 2.2. Analysis of Morpho-Physiological Traits under Saline and Alkaline Stress: A Comprehensive Examination

In this section, we delve into a detailed analysis of the correlations and associations among various morpho-physiological traits under saline and alkaline stress conditions. The analysis revealed significant associations among different traits under both alkaline and saline stresses, as shown in Figure 3 and Figure 4, highlighting the importance of understanding the interplay between these traits in the context of stress tolerance.

The SIS was significantly and negatively correlated with CHL (−0.71, −0.73), SL (−0.41, −0.34), and RL (−0.57, −0.25) for alkaline and saline stress, respectively. Conversely, the SIS was positively and significantly correlated with RNC (0.62, 0.87) under alkaline and saline stress treatments, respectively. CHL had a negative and significant correlation with the SIS (−0.71, −0.73) and RNC (−0.61, −0.55), but it was positively correlated with SL (0.46, 0.45) and RL (0.54, 0.41) under both alkaline and saline stress treatments.

Under alkaline conditions, SNC was positively and significantly correlated with RNC (0.66), SKC (0.67), and RKC (0.51), while SKC was negatively correlated with SNaK (−0.65) and RNaK (−0.2). On the other hand, few significant correlations were found among the selected traits under salinity stress, and most of them were inconsistent with the correlations observed under alkaline stress. For instance, SIS was positively correlated with RNaK, but it showed nonsignificant correlation under alkaline stress. Moreover, the SIS showed a significant positive correlation with the RNC (0.87) and RNaK (0.9) under salinity stress. On the other hand, the SIS was highly correlated with both SNC (0.83) and RNC (0.62) under alkaline stress. There was significant and positive correlation between SNC and SKC (0.73) under salinity stress, but both SNaK and RNaK were significantly negatively correlated with RKC (−0.38 and −0.42, respectively).

### 2.3. The Differential Expression of Stress-Responsive Genes under Saline and Alkaline Conditions

There were variations in the expression of the selected stress-responsive genes under both alkaline and saline stresses (Figure 5 and Figure 6). Under both stress conditions, *OsSOS1*, *OsNHX1*, and *OsIRO3* were upregulated across all genotypes. However, there was not much difference in the expression level of salt-responsive gene *OsSOS1* under both stresses, but the level of expression of salinity stress-responsive gene *OsNHX1* was higher under alkaline stress compared with saline stress in all genotypes. *OsIRO3*, an alkalinity-responsive gene, was highly expressed in all genotypes under salinity stress. Other alkalinity stress-responsive genes, *OsPPa6*, *OsFBOX335*, *OsY3IP1*, and *OsLOL5*, showed a similar pattern of expression in which expression was downregulated under both stresses in Pokkali and Mermentau, but there was downregulation under alkalinity and upregulation under salinity stress in Geumgangbyeo and Bengal. The expression pattern of the alkalinity-responsive gene *OsA6* was variable under both stresses. It was downregulated in all genotypes, except for Pokkali. However, both Pokkali and Mermentau showed an upregulation of *OsA6* under saline stress.

In the case of the salt responsive-responsive gene *OsHAK20*, upregulation was seen in Geumgangbyeo and Bengal, downregulation in Mermentau, and a contrasting expression pattern under both stresses in Pokkali. Geumgangbyeo and Bengal showed a significant increase in expression under alkalinity stress, but Pokkali and Mermentau were significantly downregulated under salinity stress. Similarly, *OsMADS25* was downregulated in Pokkali and Mermentau under saline stress but upregulated in Geumgangbyeo and Bengal. The expression level of the salt -responsive gene *OsMADS25* was the highest in Bengal under alkalinity stress followed by Geumgangbyeo, which showed a similar level of expression under both stresses. *STRK1*, a salt-responsive gene, was downregulated in Pokkali under both the stresses, but it showed upregulation in the other three genotypes, and the level of expression was higher under salinity stress compared with alkalinity stress. The expression of *OsHKT1;1* was upregulated in all genotypes under saline stress except for Pokkali, while it showed minimal expression in four different genotypes under alkaline stress.

## 3. Discussion

The present study evaluated four rice genotypes for their responses to alkaline and saline stresses based on morphological and physiological parameters. By analyzing variations in the visual stress injury scores, growth traits, ion homeostasis, chlorophyll content, and expression dynamics of stress-related genes among these genotypes, this study provided some insights into the abiotic stress tolerance mechanisms in rice plant. Due to the co-occurrence of alkaline and saline stress in coastal and arid regions, understanding the genotype-specific tolerance mechanisms can provide guidance in breeding rice varieties with improved adaptation under stressful environments. Investigating the impacts of only one type of stress, as performed in previous studies [32,33,40], limits our ability to develop varieties tolerant to multiple stresses. This study addressed this research gap through gaining an understanding of both common and distinctive adaptive mechanisms of tolerance to both stresses in rice.

The characterization of genotypes based on their resistance and sensitivity to both salt and alkali stresses is an important aspect of our study. There were significant differences in morpho-physiological parameters among rice genotypes to both stresses, which suggests variations in stress tolerance levels and adaptation mechanisms among different rice genotypes. It was possible to distinguish the genotypes based on their response to both stresses as follows: Pokkali (tolerant to salinity but sensitive to alkalinity stress), Bengal (sensitive to salinity but tolerant to alkalinity stress), Mermentau (sensitive to both salinity and alkalinity stresses), and Geumgangbyeo (tolerant to both salinity and alkali stresses). Variations in stress tolerance among different rice genotypes based on morpho-physiological parameters have been observed in previous studies [26,41,42]. Due to ionic imbalance resulting from the saline–alkali stress, reductions in chlorophyll content and root growth were observed in rice plants [43,44,45]. The genotypic differences in Na^+^ and K^+^ concentration highlighted the variability among the genotypes for the level of tolerance to alkaline and saline stresses, like earlier studies [44,46]. Internal ionic balance under both stress conditions negatively impacted crop growth [47]. Mermentau exhibited vulnerability to both stresses, as indicated by its higher SIS, lower chlorophyll content, shorter shoot and root length, and higher Na^+^/K^+^ ratio in the roots compared with other genotypes. This finding is in agreement with a previous study [48], which reported Mermentau as a sensitive rice variety to salinity stress. On the other hand, Geumgangbyeo did not show any variation in shoot Na^+^ concentration under both stresses, indicating its tolerance to both stress conditions. This observation suggests that Geumgangbyeo may have a unique mechanism for coping with both salinity and alkaline stress and the alkalinity tolerance mechanisms may be genotype-specific [49], underscoring the need for the use of multiple donors to enhance abiotic stress tolerance [44,46,50].

The observed correlations among different traits under both alkaline and saline conditions are consistent with a previous study [10]. However, the inconsistent correlations observed between the alkalinity and salinity experiments in this study may be attributed to differences in the physiological responses to both stresses, the composition, concentration of salts used in the two experiments, and the genetic background of the rice genotypes. The positive correlation between SIS and SNC, RNC, and SKC under both stress conditions was in agreement with previous studies [32,40,51], which suggests that efficient exclusion of Na^+^ from roots and shoots is necessary for enhancing stress tolerance. The negative correlation between CHL and SIS, SNC, RNC, SKC, and RNaK was also consistent with earlier studies that showed decrease in chlorophyll content and photosynthetic efficiency under salinity stress due to ion toxicity and osmotic stress [42,43]. The discrepancy in correlations between SIS and RNaK under both stresses as well as the lack of significant correlations among traits under saline stress suggests that the mechanisms of tolerance to saline and alkaline stresses differ in some aspects, which is consistent with the observation of previous study [52].

In this study, we investigated the expression of six stress-responsive genes for each stress in four genotypes under both salinity and alkalinity stresses. It is worth noting that gene expression is influenced by many factors, including the timing of stress exposure, the duration of stress exposure, the developmental stage of the plant, and the genetic background of the rice genotypes, which might have contributed to the discrepancies in gene expression and stress tolerance compared with earlier studies. Our results demonstrated the differential expression of these genes in four rice cultivars with varying levels of tolerance to salinity and alkalinity stresses. Specifically, there was downregulation of five alkalinity-responsive genes in all genotypes under alkalinity stress except *OsIRO3,* which showed upregulation in all genotypes. *OsIRO3* expression was higher under salinity stress compared with alkalinity stress in all four genotypes. Geumgangbyeo, which was tolerant to both stresses, showed a consistent pattern of reduced and increased expression for *OsY3IP1*, *OsPPa6*, and *OsLOL5* under alkalinity and salinity stress, respectively. On the other hand, Mermentau, which is sensitive to both stresses, showed reduced expression for *OsFBX335*, *OsY3IP1*, *OsPPa6*, and *OsLOL5* under both stress conditions. In case of salinity stress-responsive genes, *OsNHX1* and *OsSOS1* were upregulated in all four genotypes irrespective of the level of tolerance to both stresses whereas rest of the genes showed variable expression pattern among the genotypes. However, compared with the alkalinity-responsive genes, salinity-responsive genes showed upregulation under salinity stress conditions.

The presence of carbonate ions can cause changes in the pH and ion concentration of the growth medium, leading to altered gene expression patterns in plants exposed to this stress. Ge et al. [53] reported a downregulation of a significant number of genes in wild soybean roots in response to alkalinity stress. This suggests that carbonate stress has a tendency to turn off the expression of most genes, which contradicts our observation under salinity stress. In a study to investigate the early transcriptomic response to Na_2_CO_3_ stress in maize, Zhang et al. [54] reported altered expression of one-fourth of the genes whose expression profiles were distinct as well as common under both NaCl and high pH stresses. Our study showed significant changes in the expression of genes involved in ion homeostasis, stress response, and signaling pathways in response to saline and alkaline stresses in rice genotypes suggesting plant’s ability to alter the expression of genes involved similar biological processes under exposure to abiotic stresses. Our decision to use a lower concentration of sodium chloride (100 mM) was to keep the concentration of Na^+^ the same in both treatments for a valid comparison under both stresses. In addition, the selection of early time points for the gene expression study might have led to a lack of a distinct trend of gene expression under salinity stress, which elicited differential expression at early and late time points of exposure to salinity.

Wang et al. [22] demonstrated the role of a soluble inorganic pyrophosphatase encoding OsPPa6 in adaptation to alkaline condition in rice using CRISPR-CAS system in which knockout mutants had reduced growth and development due to low photosynthetic rate. This gene from Thellungiella halophila also enhanced alkalinity tolerance in rice [55]. In our study, although we observed downregulation of *OsPPa6* under alkaline stress in all genotypes, it was comparatively lower in alkali-tolerant genotypes, Geumgangbyeo and Bengal, than the sensitive genotypes, Pokkali and Mermentau. Both tolerant genotypes also showed upregulation under salinity stress compared with sensitive genotypes. The expression pattern of *OsLOL5*, a zinc finger protein gene with role in oxidative stress tolerance, was similar to *OsPPa6* with expression reduced under alkaline stress in alkalinity-tolerant genotypes compared with susceptible genotypes. This result was contrary to [39], who showed improved tolerance to saline–alkaline stress in *OsLOL5* overexpressing Arabidopsis, yeast, and rice due to increased accumulation of proline and soluble sugar in transgenic plants under stress conditions.

In an overlapping major QTL for alkali tolerance score, shoot Na^+^ concentration, and Na^+^/K^+^ ratio of shoots, Li et al. [36] identified *OsIRO3*, a bHLH-type transcription factor, as a negative regulator of the Fe-deficiency response in rice. A 7-bp insertion/deletion (indel) distinguished alkalinity-tolerant from the alkalinity-sensitive rice varieties. Previous studies reported involvement of *OsIRO3* in response to drought, alkalinity, and salinity stresses [36,56,57]. Interestingly, upregulation of *OsIRO3* in response to both stresses in our study suggest that it may have a multifaceted role in the regulation of stress responses beyond iron deficiency in rice.

The *OsY3IP1* is a nucleus-encoded thylakoid protein with an important role in the assembly of photosystem I and, thus, is involved in minimizing photooxidative damage under stress conditions. It was downregulated under alkaline stress in all genotypes, but the magnitude was much lower in alkalinity-tolerant Geumgangbyeo and Bengal compared with susceptible genotypes. On the other hand, it was upregulated under salinity stress in Geumgangbyeo and Bengal. This expression pattern, which was similar to *OsPPa6* and *OsLOL5*, was contradictory to the finding of Moon et al. [23], who showed increased tolerance to salinity and alkalinity stress in overexpressing rice plants due to reduced reactive oxygen species accumulation. Both *OsA6* and *OsFBX335* were identified from the alkalinity tolerance QTL region based on sequence variation distinguishing alkalinity-tolerant Cocodrie from sensitive N22 [32]. The reduced expression of this gene in our study was in agreement with their result, suggesting it is a negative regulator of alkali tolerance. This observation was further supported by earlier studies [37,38].

Na^+^/K^+^ homeostasis in rice is crucial for tolerance to both salinity and alkalinity stresses. In addition to the known genes (*OsHKT1:1*, *OsNHX1*, and *OsSOS1*), expression of three more genes (*OsHAK20*, *STRK1*, and *OsMADS25*), which were potential candidates located in the salt tolerance QTL intervals [28], was analyzed. These were responsive to salinity stress at the seedling stage and showed differential expression in both salt-tolerant and -susceptible genotypes. *OsSOS1* is responsible for transporting Na^+^ ions from the cytoplasm to the extracellular space, thereby facilitating Na^+^ exclusion from the cytoplasm and contributing to salt tolerance in rice [13,58]. The significant increase in *OsSOS1* expression across all cultivars in response to both alkaline and saline stress conditions observed in our study was in agreement with a previous study [59]. But Farooq et al. [26] reported downregulation of *OsSOS1* in Pokkali shoots 24 h after saline stress. The discrepancy in *OsSOS1* expression between this study and earlier studies could be due to different experimental conditions like stress exposure time, intensity of stress, and type of stress. Short-term stresses may elicit an initial increase in this transporter that helps regulate ion balance, whereas longer or more severe disruptions could activate alternative pathways or coordination of *OsSOS1* with other genes.

*OsNHX1*, which confers salt tolerance in plants by vacuolar sequestration of Na^+^ ions [60], was highly upregulated under alkalinity stress compared with salinity stress in all genotypes, which was consistent with previous studies [26,61,62]. It is possible that this upregulation is related to the very high Na^+^ load and higher Na^+^/K^+^ ratio observed in leaves may trigger the upregulation of genes such as *OsSOS1* and *NHX1* for Na^+^ exclusion/sequestration and/or K^+^ accumulation as a damage control measure, especially under alkaline stress.

Under salt stress, competition between Na^+^ and K^+^ ions at K^+^ binding sites (Maathuis and Amtmann 1999) or increased cytosolic Na^+^/K^+^ ratio [63], could lead to K^+^ deficiency and, consequently, the high induction of *OsHKT1* in rice genotypes occurs as in the present study. The variation in stress level and duration could account for the reduced expression of *OsHKT1;1* in salt-tolerant Pokkali [26]. Kader et al. [64] reported that *OsHKT1;1* expression was significantly reduced in Pokkali when exposed to 150 mM salt stress, compared to salt-sensitive BRRI Dhan29. This finding aligns partially with our results. The upregulation of *OsHKT1;1* under saline stress is consistent with previous studies, [13,15]. The role of *OsHKT1;1* in preventing Na^+^ accumulation in shoot in response to salt stress and mediation of alkali cation transport was reported in earlier studies [15,65]. A short treatment period of 6 h might be insufficient to significantly disrupt homeostasis and induce this transporter, particularly in tolerant genotypes like Pokkali. Long-term ionic imbalance is probably needed to activate adaptive mechanisms like increased *OsHKT1* expression. Mild or brief stresses only moderately impact physiology and metabolism, limiting gene expression changes. Although sensitive varieties have less robust regulation of ion homeostasis, *OsHKT1* induction may occur early upon exposure to moderate level of stress. Tolerant genotypes with stronger regulatory mechanism can maintain it without immediately increasing this transporter through alternative mechanisms or slower kinetics. It is possible that the concentration of salt stress used in our study was probably not high enough to induce gene expression in certain varieties [64].

*OsMADS25* enhanced salinity tolerance by increasing accumulation of osmoprotectants and ROS scavenging capacity [29]. In contrast to its upregulation in salinity-tolerant TCCP and FL478 [28], we noticed enhanced expression in salinity- and alkalinity-tolerant Geumgangbyeo, as well as in Bengal, which is tolerant to alkalinity but susceptible to salinity stress. It was highly downregulated in Pokkali and Mermentau with contrasting response to salt stress. *OsHAK20* is a high-affinity potassium transporter with a potential role in maintaining Na^+^/K^+^ homeostasis under salt stress and was highly expressed in salt-tolerant TCCP and FL478 compared with salt-susceptible Jupiter [28]. However, our findings did not show any trend differentiating tolerant from susceptible genotypes to either stress. *OsSTRK1* is a receptor-like cytoplasmic kinase that enhances rice plant’s tolerance to salinity and oxidative stress at both seedling and flowering stages [27]. This gene showed upregulation in response to both stresses in Geumgangbyeo, Mermentau, and Bengal, but its expression was higher under salt stress compared with alkalinity stress. Its expression was reduced more under salt stress compared with alkali stress in Pokkali. The expression pattern was in contrast to the results of [28]. In summary, this study investigated the gene expression patterns of various genes under saline and alkaline stress in different rice genotypes. The results revealed that the expression of several genes, including *OsPPa6*, *OsLOL5*, *OsIRO3*, *OsY3IP1*, *OsA6*, *OsFBX335*, *OsHKT1;1*, *OsNHX1*, and *OsMADS25*, varied in response to the different stress conditions. The study found that there were differences in the expression patterns of these genes between tolerant and susceptible genotypes. For example, the expression of *OsPPa6* and *OsLOL5* was comparatively lower in alkali-tolerant genotypes than in sensitive genotypes under alkaline stress. Conversely, the expression of *OsNHX1* was highly upregulated under alkaline stress compared with salinity stress in all genotypes. The study also discussed the potential implications of the gene expression patterns for salt and alkaline stress tolerance in rice, highlighting the importance of understanding the interplay between different genes and their responses to stress conditions.

## 4. Materials and Methods

### 4.1. Plant Materials

The four rice genotypes Pokkali, Geumgangbyeo, Bengal, and Mermentau, with variable tolerance to salinity stress, were selected for this investigation. Bengal and Mermentau are sensitive to salinity stress, whereas Pokkali and Geumgangbyeo are highly tolerant and tolerant to salinity, respectively [66]. Bengal and Mermentau are high-yielding varieties released by the Louisiana State University Agricultural Center [67,68].

### 4.2. Evaluation of Salinity and Alkalinity Tolerance

The salinity and alkalinity tolerance screening experiments were conducted at the Louisiana State University Agricultural Center greenhouse in Baton Rouge, LA, USA (30°24′41.7″ N, 91°10′21.8″ W). The temperature range in the greenhouse was maintained at 25–29 °C. The detailed screening procedure included three steps: (a) pre-germination of seeds, (b) stress exposure, and (c) data collection. The seeds were surface-sterilized with a 5% (*v*/*v*) sodium hypochlorite solution for 30 min followed by thorough washing with distilled water and transferred to Petri dishes containing filter paper. The pre-germinated seeds were then transferred to a hydroponic system containing nutrient solution which comprised of 0.1% Jack’s professional fertilizer (J.R. Peters, Inc., Allentown, PA, USA) and Peters professional liquid S.T.E.M. Supplement (Everris Na Inc. Dublin, OH, USA) (12.5 mL per 10 L). The hydroponic experiment was carried out in triplicate. Seedlings at the three-leaf stage were exposed to alkaline and saline stresses of 50 mM Na_2_CO_3_ at pH 10.0 and 100 mM NaCl at pH 5.0, respectively. Treatment was continued for 4 and 6 days in the case of alkalinity and salinity treatments, respectively. Control plants were grown in nutrient solutions without any treatment. A factorial design with three replications was followed. Ten seedlings were grown for each genotype, but five plants with uniform growth per genotype were selected for data collection in each replication. The trait means in each genotype and each replication were obtained by averaging the measured values of five seedlings. Observations were taken on the following traits.

### 4.3. Visual Salt Injury Score (SIS)

To assess the plant’s response to salinity stress, a visual salt injury score (SIS) was determined 5 days post-salinization (DPS) using the IRRI standard evaluation system [69]. The SIS was recorded on a scale of 1–9, where a score of 1 indicated no injury, a score of 3 was assigned to seedlings showing little leaf damage but were stunted compared to the control, a score of 5 represented stressed plants with stunted growth, green rolled leaves, and whitish tips, a score of 7 indicated a plant with only a green stem and dried leaves, and a score of 9 was assigned to a completely dead plant. The SIS values were based on the observation of 10 seeds for each genotype in three replicates, and the final scores were calculated by taking the mean of the scores of all the seeds.

### 4.4. Chlorophyll Content (SPAD Reading)

The SPAD 502 chlorophyll meter (Spectrum Technologies, Inc. Aurora, IL, USA) was used to determine the relative chlorophyll content in the mid-part of the second youngest leaf of control and stressed rice genotypes 4 days after exposure to salinity stress. In case of alkalinity stress, observations were taken on the 3rd day after the imposition of stress.

### 4.5. Growth Parameters

Growth parameters were assessed by measuring the shoot and root length of each genotype 6 days after exposure to stress. The shoot length was measured from the base of the plant to the tip of the longest leaf, while the root length was measured from the base of the plant to the tip of the root mass.

### 4.6. Measurement of Na^+^ and K^+^

The concentrations of Na^+^ and K^+^ in the roots and shoots of each genotype were determined following the method of [70]. Plant samples of each genotype were dried at 65 °C for 2 days and then ground to fine powder using a mortar and pestle. Approximately 500 mg of plant shoot and 100 mg of root tissue were digested with 5 mL of nitric acid and 3 mL of hydrogen peroxide at a temperature range of 152–155 °C for 3 h. The digested tissue was then diluted to a final volume of 12.5 mL. The concentration of Na^+^ and K^+^ in each sample was determined using a flame photometer (Jenway PFP7 model; Bibby Scientific Ltd., Staffordshire, UK). The final concentrations of Na^+^ and K^+^ were determined using a standard curve generated from various dilutions of Na^+^ and K^+.^ The Na^+^/K^+^ ratio of the root and shoot was calculated by dividing the concentration of Na^+^ by the concentration of K^+^.

### 4.7. Expression Analysis of Selected Genes

Quantitative real-time reverse transcription PCR (qRT-PCR) was used to measure the expression of selected genes in rice genotypes under control, saline, and alkaline stress conditions. The leaf samples were collected at two time points (0 and 6 h after stress exposure) and were immediately frozen in liquid nitrogen and stored at −80 °C. Three biological replicates per treatment were used for total RNA extraction using Trizol reagent. RNA quality was assessed using 1.2% agarose gel and RNA quantification was performed using an ND-1000 Spectrophotometer (Thermofisher Scientific, Waltham, MA, USA). RNA samples were treated with PerfeCTa DNase 1 (Quantabio, Beverly, MA, USA) and the resulting high-quality RNA was reverse transcribed into cDNA using the iScipt™ first strand cDNA synthesis kit (Bio-Rad Laboratories, Hercules, CA, USA).

The sequences of stress-responsive genes were obtained from the Phytozome database [71], and qRT-PCR primers were designed using Primer Quest (Integrated DNA Technologies, Coralville, IA, USA) (Table 3). EF1α (LOC_Os03g08010) was used as an internal standard for expression normalization. The qRT-PCRs were conducted in three technical replicates using cDNA from the biological replicates and iTaq™ Universal SYBR Green Supermix (Bio-Rad Laboratories, Hercules, CA, USA) [72]. The expression levels of the genes were determined using the 2^−∆∆CT^ method [73].

### 4.8. Statistical Analysis

The statistical analysis was conducted in R version 4.3.2 [74]. Since the treatment levels for two main effects were cross-classified with each other, factorial treatment arrangement with a complete randomized design was used for data analysis. The Aov function in R was used to conduct analysis of variance (ANOVA). For post hoc comparisons, a Tukey test was performed for determining significance of the main effects. Boxplot, descriptive statistics, and Pearson correlation coefficients for each trait under different stress treatments were obtained using R.

## 5. Conclusions

Our study focused on the morpho-physiological and molecular responses of four rice genotypes (Geumgangbyeo, Pokkali, Bengal, and Mermentau) to salinity and alkalinity stresses, which are becoming increasingly prevalent due to climate change. These stresses negatively impact rice production worldwide. Our research aimed to provide insights into the differential tolerance of these genotypes to salinity and alkalinity stresses, as well as the molecular mechanisms underlying their response to these stresses. The results indicated that Geumgangbyeo was the most tolerant, while Mermentau was the least tolerant to both stresses. Pokkali and Bengal were tolerant only to salinity and alkalinity stresses, respectively. A set of stress-responsive genes showed differential expression in these rice genotypes under both stress conditions, and their expression patterns were consistent with the observed morphological responses. Our study suggests that the divergence in response to alkalinity and salinity stresses among rice genotypes could be due to different molecular mechanisms conferring tolerance to each stress. Further investigations are needed to differentiate the molecular bases underlying tolerance to both stresses. In conclusion, our research highlights the importance of understanding the molecular and physiological mechanisms underlying tolerance to salinity and alkalinity stresses in rice. By identifying genes and pathways differentially expressed under stress conditions, we can potentially contribute to the development of rice varieties with improved stress tolerance, thereby contributing to global food security and sustainability.

## Figures and Tables

**Figure 1 plants-13-00060-f001:**
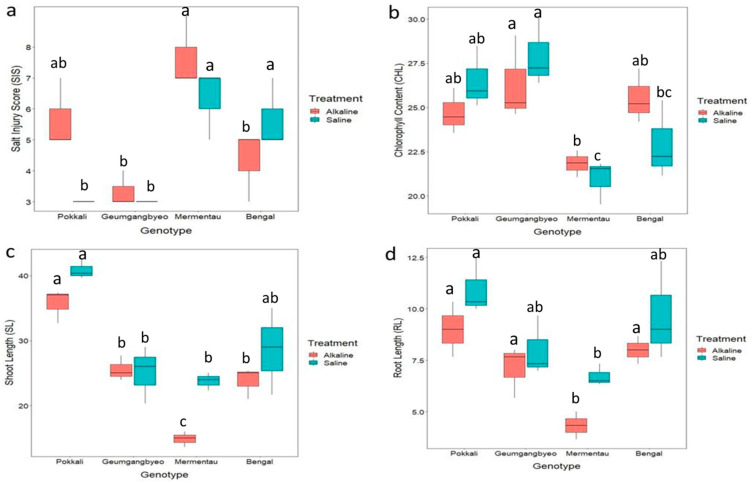
Response of four different genotypes (Bengal, Geumgangbyeo, Mermentau, Pokkali) under alkaline and saline stress. (**a**) SIS, salt injury score; (**b**) CHL, chlorophyll content (SPAD units); (**c**) SL, shoot length (cm); (**d**) RL, root length (cm). a, b, c represent Tukey lettering for determining difference among genotypes, and shared letters between the genotypes indicate no significant difference at 0.05 probability level.

**Figure 2 plants-13-00060-f002:**
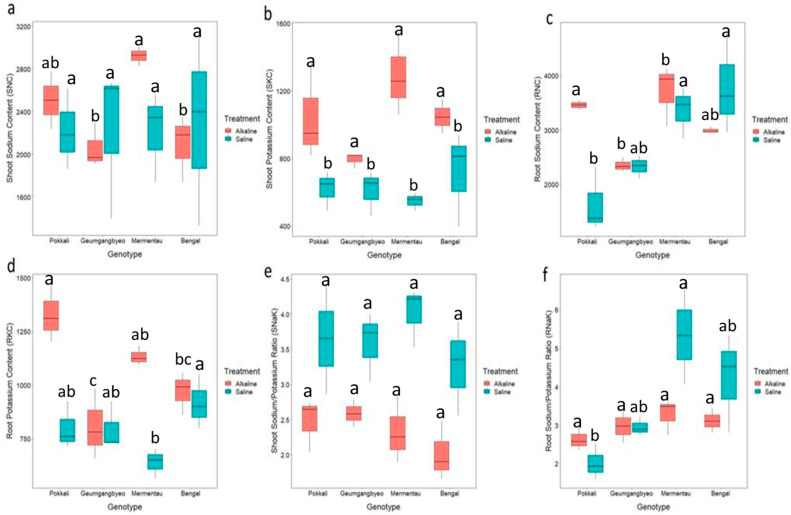
Response of four different genotypes (Bengal, Geumgangbyeo, Mermentau, Pokkali) under alkaline and saline stress. (**a**) SNC, shoot sodium concentration (mmol kg^−1^); (**b**) SKC, shoot potassium concentration (mmol kg^−1^); (**c**) RNC, root sodium concentration (mmol kg^−1^); (**d**) RKC, root potassium concentration (mmol kg^−1^); (**e**) SNaK, ratio of the shoot sodium and potassium concentration (ratio); (**f**) RNaK, ratio of the root sodium and potassium concentration (ratio). a, b, c represent Tukey lettering for determining difference among genotypes, and shared letters between the genotypes indicate no significant difference at 0.05 probability level.

**Figure 3 plants-13-00060-f003:**
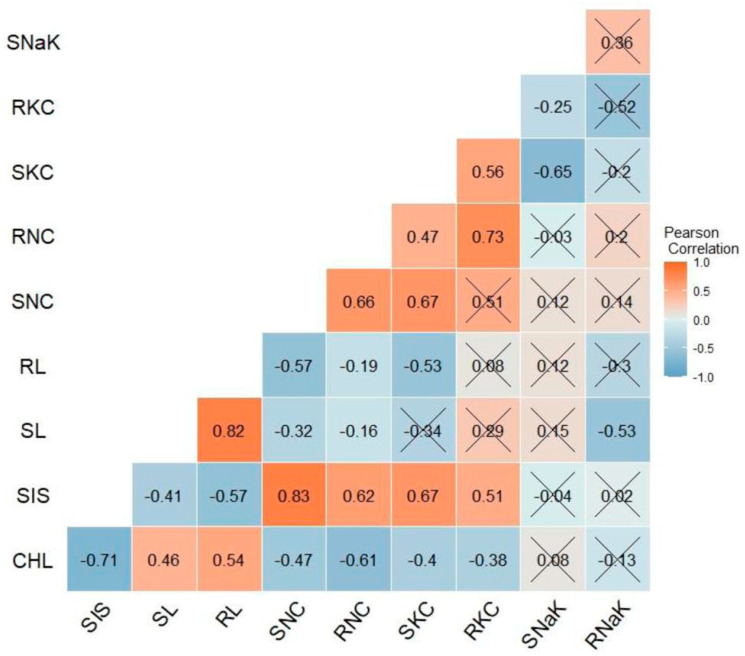
Pearson correlation matrix of different morpho-physiological traits at the seedling stage under alkaline stress. X depicts nonsignificant correlation between the traits at probability level of 0.05. SIS, salt injury score; CHL, chlorophyll content, SL, shoot length; RL, root length; SNC, shoot sodium concentration, RNC, root sodium concentration; SKC, shoot potassium concentration; RKC, root potassium concentration, SNaK, ratio of the shoot sodium and potassium concentration; RNaK, ratio of the root sodium and potassium concentration.

**Figure 4 plants-13-00060-f004:**
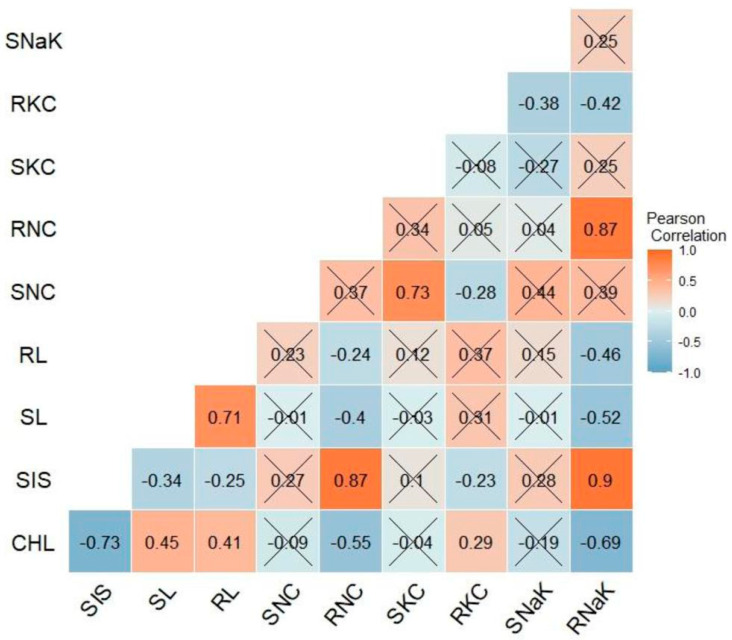
Pearson correlation matrix of different morpho-physiological traits at the seedling stage under saline stress. X depicts nonsignificant correlation between the traits at probability level of 0.05. SIS, salt injury score; CHL, chlorophyll content, SL, shoot length; RL, root length; SNC, shoot sodium concentration, RNC, root sodium concentration; SKC, shoot potassium concentration; RKC, root potassium concentration, SNaK, ratio of the shoot sodium and potassium concentration; RNaK, ratio of the root sodium and potassium concentration.

**Figure 5 plants-13-00060-f005:**
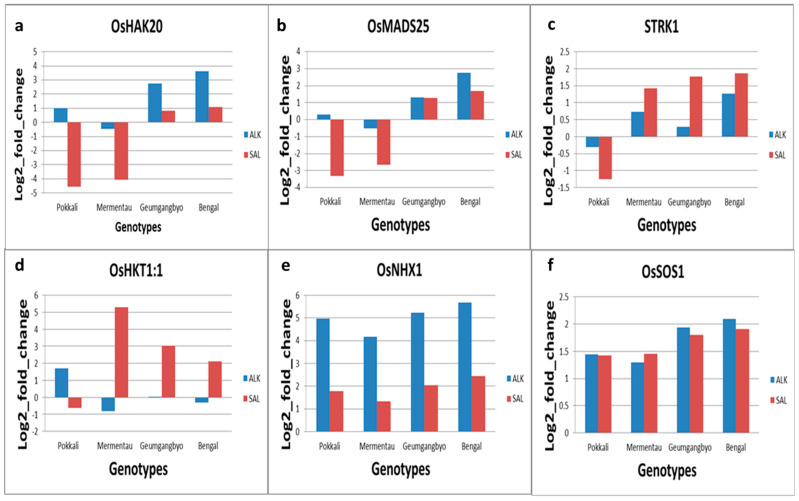
Expression analysis of selected salinity stress-responsive genes 6 h after imposition of saline and alkaline stress in Pokkali (Pok), Geumgangbyeo (Geu), Mermentau (Mer), and Bengal (Ben). The y-axis represents the Log2 fold change in mRNA expression compared with control (no stress), while x-axis represents the genotypes. The genes are (**a**) *OsHAK20* (LOC_Os02g31940), (**b**) *OsMADS25* (LOC_Os04g23910), (**c**) *STRK1* (LOC_Os04g45730), (**d**) *OsHKT1;1* (LOC_Os06g48810), (**e**) *OsNHX1* (LOC_Os07g47100), and (**f**) *OsSOS1* (LOC_Os12g44360).

**Figure 6 plants-13-00060-f006:**
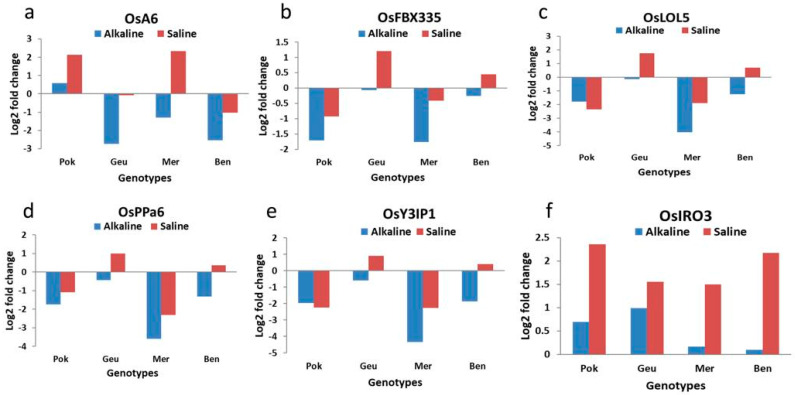
Expression analysis of selected alkalinity stress-responsive genes 6 h after imposition of saline and alkaline stress in Pokkali (Pok), Geumgangbyeo (Geu), Mermentau (Mer), and Bengal (Ben). The y-axis represents the Log2 fold change in mRNA expression compared with control (no stress), while x-axis represents the genotypes. The genes are (**a**) *OsA6* (LOC_Os09g32550), (**b**) *OsFBX335* (LOC_Os09g32860), (**c**) *OsLOL5* (LOC_Os01g42710.1), (**d**) *OsPPa6* (LOC_Os02g52940), (**e**) *OsY3IP1* (LOC_Os01g58470.1), and (**f**) *OsIRO3* (LOC_Os03g26210).

**Table 1 plants-13-00060-t001:** Trait means for ten morpho-physiological parameters in rice genotypes at the seedling stage under saline stress.

Traits ^$^	Pokkali	Geumgangbyeo	Mermentau	Bengal	Trait Mean
SIS	3.0 ^b^ ± 0.0	3.0 ^b^ ± 0.0	7.0 ^a^ ± 1.15	6.0 ^a^ ± 1.15	4.8
CHL (SPAD units)	26.5 ^ab^ ± 01.7	27.9 ^a^ ± 1.95	21.0 ^c^ ± 1.23	22.9 ^bc^ ± 2.20	24.6
SL (cm)	40.9 ^a^ ± 2.06	25.1 ^b^ ± 4.12	23.8 ^b^ ± 1.88	28.6 ^b^ ± 3.18	29.6
RL (cm)	10.9 ^a^ ± 1.35	8.5 ^ab^ ± 1.45	6.7 ^b^ ± 0.53	9.7 ^ab^ ± 1.40	9
SNC (mmol kg^−1^)	2218 ^a^ ± 108.35	2227 ^a^ ± 116.59	2214 ^a^ ± 106.01	2294 ^a^ ± 114.23	2238
SKC (mmol kg^−1^)	619 ^b^ ± 114.3	685 ^b^ ± 132.68	548 ^b^ ± 51.40	715 ^b^ ± 82.36	642
RNC (mmol kg^−1^)	1640 ^b^ ± 91.53	2333 ^ab^ ± 108.39	3365 ^a^ ± 166.32	3793 ^a^ ± 128.07	2783
RKC (mmol kg^−1^)	799 ^ab^ ± 88.96	795 ^ab^ ±90.04	639 ^b^ ± 68.34	917 ^a^ ± 98.94	788
SNaK (ratio)	3.7 ^a^ ±0.78	3.6 ^a^ ±0.49	4.0 ^a^ ±0.42	3.3 ^a^ ± 0.66	3.6
RNaK (ratio)	2.2 ^b^ ± 0.45	3.0 ^ab^ ± 0.23	5.4 ^a^ ± 0.29	4.2 ^ab^ ± 0.27	3.7

**^$^** SIS, salt injury score; CHL, chlorophyll content, SL, shoot length; RL, root length; SNC, shoot sodium concentration; SKC, shoot potassium concentration, RNC, root sodium concentration; RKC, shoot potassium concentration; SNaK, ratio of sodium and potassium concentration in shoot; RNaK, ratio of sodium and potassium concentration in root. Trait means are averages of three replications. ^a, b, c^ represent Tukey lettering for determining difference among genotypes, and shared letters between the genotypes indicate no significant difference at 0.05 probability level.

**Table 2 plants-13-00060-t002:** Trait means for ten morpho-physiological parameters in rice genotypes at the seedling stage under alkaline stress.

Traits ^$^	Pokkali	Geumgangbyeo	Mermentau	Bengal	Trait Mean
SIS	5.7 ^ab^ ± 1.15	3.3 ^b^ ± 0.57	7.7 ^a^ ± 1.15	4.3 ^b^ ± 1.15	5.3
CHL (SPAD units)	4.7 ^ab^ ± 1.28	26.3 ^a^ ± 1.39	21.8 ^b^ ± 0.75	25.5 ^ab^ ± 1.52	24.6
SL (cm)	35.7 ^a^ ± 2.29	25.6 ^b^ ± 1.82	14.9 ^c^ ± 0.94	23.8 ^b^ ± 0.23	25
RL (cm)	9.0 ^a^ ± 1.3	7.1 ^a^ ± 0.87	4.3 ^b^ ± 0.66	8.0 ^a^ ± 0.66	7.1
SNC (mmol kg^−1^)	2506 ^ab^ ± 130.34	2057 ^b^ ± 100.94	2923 ^a^ ± 45.00	2088 ^b^ ± 113.89	2394
SKC (mmol kg^−1^)	1044 ^a^ ±140.51	794 ^a^ ± 45.4	1289 ^a^ ± 123.38	1047 ^a^ ± 101.42	1044
RNC (mmol kg^−1^)	3458 ^a^ ± 49.78	2357 ^b^ ± 87.53	3708 ^b^ ± 111.68	3000 ^ab^ ±92.20	3254
RKC (mmol kg^−1^)	1328 ^a^ ± 68.20	808 ^c^ ± 80.81	1134 ^ab^ ± 22.68	970 ^bc^ ± 77.48	1245
SNaK (ratio)	2.47 ^a^ ± 0.38	2.59 ^a^ ± 0.19	2.33 ^a^ ± 0.47	2.01 ^a^ ± 0.41	2.4
RNaK (ratio)	2.6 ^a^ ± 0.30	3.0 ^a^ ± 0.43	3.3 ^a^ ± 0.17	3.1 ^a^ ± 0.31	3

**^$^** SIS, salt injury score; CHL, chlorophyll content, SL, shoot length; RL, root length; SNC, shoot sodium concentration; SKC, shoot potassium concentration, RNC, root sodium concentration; RKC, shoot potassium concentration; SNaK, ratio of sodium and potassium concentration in shoot; RNaK, ratio of sodium and potassium concentration in root. Trait means are averages of three replications. ^a, b, c^ represent Tukey lettering for determining difference among genotypes, and shared letters between the genotypes indicate no significant difference at 0.05 probability level.

**Table 3 plants-13-00060-t003:** Primers for real time q-PCR study for salt and alkaline stress.

Genes	MSU ID	Forward Primer (5′–3′)	Reverse Primer (5′–3′)
*OsHKT1;1*	LOC_Os06g48810	GGCGTTTCTGGCATCAACTGTC	ATTCCAGTCGACAGCACCGAAC
*OsNHX1*	LOC_Os07g47100	TGACCGTGAGGTTGCCCTTATG	GAGAATGCCGCTCAAATCTAGCAA
*OsSOS1*	LOC_Os12g44360	AGATCGCGCTTACTCTTGCTGTC	AGACCTCCAGTGCATCTTGTGC
*OsHAK20*	LOC_Os02g31940	CGAGGGTTGGTGTACCTGAT	GGTTTTTCCTCAAGCGAGTG
*OsMADS25*	LOC_Os04g23910	CTCTGGAGAAAGCACGTCAA	GACTCAATTCAAGGTCAATACACAC
*STRK1*	LOC_Os04g45730	CCTCGACGCCAACATGAA	TGAGGTGTGGGTCTACGTATC
*OsA6*	LOC_Os09g32550	CGAGAAGCTGAACGAGACG	CGAGTTGAAGGCGTAGCTG
*OsFBX335*	LOC_Os09g32860	CAGTGCCTAGCCTTCCAGAG	TGACGAAAAGCACGAGACAC
*OsLOL5*	LOC_Os01g42710.1	GCAACCCACAAGAACTAACTCATC	GGCTTGTCCATACCATCTTGAAC
*OsPPa6*	LOC_Os02g52940	TGAGCTTGACTGGAAAATTGTG	GCTTCTCAACATCATCCACATC
*OsY3IP1*	LOC_Os01g58470.1	CCAGGTCAAAAGGGTGCTTG	TCTCCTTCGCAAGCAACTGA
*OsIRO3*	LOC_Os03g26210	TGGTCGATTGGTTTTCAGCAG	AACCTTCCTCGGGACCTTCT
*OsEF1α*	LOC_Os03g08010	TTGATCTGGTCAAGAGCCTCAAGC	TCTCTGGGTTTGAGGGTGACAACA

## Data Availability

The study did not report any data. All data, tables and figures in this manuscript are original.

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
