# Peer review of "Comparison of the Morpho-Physiological and Molecular Responses to Salinity and Alkalinity Stresses in Rice"

_plants, 2023, doi:10.3390/plants13010060_

Round 1
Reviewer 1 Report
Comments and Suggestions for Authors
The research conducted a comparative analysis of multiple phenotypes and gene expressions among four rice genotypes under salt and alkalinity stress conditions. The outcomes revealed distinct responses of various rice genotypes to these two abiotic stressors at different levels, offering crucial insights for further exploration of the molecular mechanisms governing plant responses to closely related abiotic stress conditions. The manuscript demonstrates adept writing and organization. One significant concern pertains to how the authors conclude the differences in tolerance among these genotypes to the two stressors. Given that the primary conclusion underscores Geumgangbyeo as the most tolerant and Mermentau as the least tolerant to both stresses, the manuscript should articulate the evidence supporting this conclusion accurately across distinct sections. Other minor revisions are noted below.
Line 59. Please rectify the reference error.
Line 95. Correct [31-33) to [31-33].
Lines 130-133. Clarify the meaning of "SIS" and provide its calculation method, including the explanation of all abbreviations when they first appear.
Figure 1 and 2. Add statistical results of trait changes among genotypes and treatments to Figures 1 and 2, along with the corresponding units for each character.
Line 165. Corrected the citation to Figure 1.
Line 167. “Figure 2.”
Table 1 and 2. Include SD values of each trait in Tables 1 and 2. Did the authors determine the changes in these traits under control condition? I cannot understand how to calculate the treatment effect on the shifts in these characters of the plant without the data under control conditions. Since the results in Tables 1 and 2 are replicate with Figure 1 and 2, I suggest the authors delete both Tables 1 and 2, and move Table S1 to the manuscript.
Lines 192-196. Please provide statistical results in the text when describing relationships as "significant".
Lines 222-257. Gene names should be italic. And why these genes were selected? According to these results, there is a control treatment in the present study. So, please present the exact value or relative values of each trait in the text.
Figure 3C. The legend of the Y-axis in this panel is missing.
Lines 282-286. Please refer to specific results explaining changes in the tolerance of genotypes to different stresses.
Line 316. Please correct the sentence to “the observation of previous study”.
Lines 475-481. Did the authors mean that the SIS values of each genotype were estimated by observation the changes in the phenotype of the seedlings after a 5-day treatment? How many seedlings were examined in the test (5?)? And how to exactly define the scores of 6 or 8?
Reviewer 2 Report
Comments and Suggestions for Authors
In the manuscript (Comparison of the Morpho-Physiological and Molecular Responses to Salinity and Alkalinity Stresses in Rice), the authors examined the morpho-physiological and molecular response of four rice genotypes to both salinity and alkalinity stresses. Overall, this study suggested that divergence in response to alkalinity and salinity stresses among rice genotypes could be due to different molecular mechanisms conferring tolerance to each stress. However, I think that the manuscript should be revised before publication.
Major:
1. In the result part of the article, the author's analysis of the chart is not accurate enough. For example, in line 164, I think it should be "There is no significant difference in the shoot length in Geumgangbyeo under both alkaline and saline stresses ”.
2. The subheading is not accurate enough. Example 2.2 Lack of subject.
3. In the discussion section, the authors paid more attention on the expression of different types of genes in rice. There is a lack of summary at the end.
Minor:
There are some minor errors in the format of the thesis.
1. Line 59: The quoted article should not be placed after “likewise”.
2. Line 74: The cited article number should be followed by a comma by a space. The same problem occurs in lines 75, 274, 291, 301, 308, 407, and 421.
3. Line 92: There should be a space after the period. The same problem occurs in line 98, line 343, line 354, line 398 and line 446.
4. Line 95: Brackets should be replaced with square brackets.
5. Line 138: The comma after "and" should come before "and".
6. Line 295: One square bracket is missing.
7. Line 303: There's an extra period.
8. Line 343: “Na2CO3” should be changed to “Na2CO3”.
9. Line 406: There is an extra space after "[69]".
10. Line 446: Added "4. Materials and".
11. Line 167: "SNaK" should be changed to "RNaK".
Author Response
"Please see the attachment."

Reviewer 3 Report
Comments and Suggestions for Authors
In the research work, the authors evaluated the morpho-physiological response of four rice genotypes exhibiting diversity in their response to salinity and alkaline tolerance. After carefully evaluating the manuscript, I feel that the work does not contribute novelty to the existing knowledge. A hypothesis-driven testing or robust analysis is lacking in the work, and the present form is merely a simple analysis of the four genotypes with some qPCR gene validation. The manuscript could have improved in many ways, for instance, RNASeq study of these genotypes may lead to the discovery of many genes.
Tables 1 and 2: How do these genotypes perform under control? When you say treatment effect, please include the controls.
Line 137-139:” On the con-137 tray, no significant difference was observed for SIS in Geumgangbyeo, Bengal, and, Mermentau under alkaline conditions compared to saline stress (Figure 1).” Table 2 shows that SIS in Mermentau is statistically different from Geumgangbyeo and Bengal. Also, SIS in Bengal between the two treatments is different. Figures 1 and 2 are shown without any statistical support.
Line 223: Italicize the gene names.
Line 129: Expand SIS, SL and RL etc. at the first cite.
Comments on the Quality of English LanguageModerate editing will improve the manuscript's readability.
Author Response
"Please see the attachment."

Round 2
Reviewer 3 Report
Comments and Suggestions for Authors
The authors have revised the manuscript and addressed most of the comments.